# Innovations in Chronic Pain Treatment: A Narrative Review on the Role of Cryoneurolysis

**DOI:** 10.3390/medicina61061090

**Published:** 2025-06-15

**Authors:** Andrea Tinnirello, Maurizio Marchesini, Silvia Mazzoleni, Carola Santi, Giuliano Lo Bianco

**Affiliations:** 1Anesthesia and Pain Management Unit, ASST Franciacorta, Iseo Hospital, 25049 Brescia, Italy; andrea.tinnirello@asst-franciacorta.it; 2Department of Anesthesia and Pain Medicine, Mater Olbia Hospital, 07026 Olbia, Italy; marchesinidoc@gmail.com; 3Pain Management Unit, IRRCS Fondazione Maugeri Pavia, 27100 Pavia, Italy; silvia.mazzoleni09@gmail.com; 4Anesthesia Intensive Care and Pain Management Unit, Manerbio Hospital, ASST Garda, 25025 Brescia, Italy; carola.g.santi@gmail.com; 5Anesthesiology and Pain Department, Foundation G. Giglio Cefalù, Via Pietrapollastra, Cefalù, 90015 Palermo, Italy

**Keywords:** chronic pain, neuromodulation, spinal cord stimulation, pulsed radiofrequency, peripheral nerve stimulation, deep brain stimulation, pain management, transcutaneous electrical nerve stimulation, neuroplasticity, interventional pain therapy

## Abstract

*Background and Objectives:* Chronic pain is a significant global health issue, with conventional treatment strategies often proving insufficient or causing undesirable side effects. Interventional pain management techniques, including neuromodulation, have gained increasing interest as alternative therapeutic options. Cryoneurolysis, a technique leveraging extreme cold to modulate pain pathways, has emerged as a promising tool in chronic pain management. However, its efficacy and role within current clinical practice remain under evaluation. *Methods*: A narrative review was conducted by searching PubMed, Scopus, Embase, and Web of Science databases for studies published between 2010 and 2024 using the keywords “Cryoneurolysis”, “Cryoanalgesia”, “Cryoablation”, and “Chronic pain.” Only English-language studies were included. Studies that examined intraoperative cryoablation or lacked statistical analyses (except case reports) were excluded. *Results*: A total of 55 studies were included: 4 randomized controlled trials (RCTs), 16 retrospective studies, 4 prospective observational studies, and 31 case reports or small case series. The studies displayed significant heterogeneity in patient selection, targeted nerves, procedural protocols, and follow-up durations. While two RCTs demonstrated a significant pain reduction compared to control groups, other RCTs reported no significant improvement. Observational studies and case reports frequently report positive outcomes, with some achieving complete pain relief. Cryoneurolysis appears to be most effective in treating neuropathic pain, particularly in patients with peripheral nerve involvement. *Conclusions:* Cryoneurolysis is a safe technique for chronic pain management, which has been successfully applied, particularly for selected neuropathic pain conditions. However, the current evidence is limited by study heterogeneity and a lack of high-quality comparative trials. Further well-designed randomized studies are necessary to define its long-term efficacy and its potential role relative to other interventional pain therapies, such as radiofrequency ablation.

## 1. Introduction

Chronic pain is a global health concern that affects millions of people and imposes a significant burden on healthcare systems worldwide [1,2]. According to the International Association for the Study of Pain (IASP), it is defined as pain that persists or recurs for more than three months, often leading to a host of psychosocial, functional, and economic challenges [1,3]. Despite advancements in conventional medical management—including pharmacological interventions, physical therapy, and psychological support—many patients continue to experience intractable pain or are hindered by the side effects of long-term medication use [4,5]. These limitations have spurred interest in interventional pain therapies and neuromodulation techniques, which target specific pathways within the nervous system to modulate nociceptive signal transmission. Within the broader field of neuromodulation, a variety of approaches have been investigated for their ability to provide safe and sustained pain relief while minimizing adverse effects. Spinal cord stimulation (SCS), peripheral nerve stimulation (PNS), dorsal root ganglion (DRG) stimulation, deep brain stimulation (DBS), and motor cortex stimulation (MCS) are among the better-known therapies; however, novel modalities continue to emerge [6]. One of the most intriguing recent advances is cryoneurolysis (or cryoanalgesia), which leverages the application of ultra-low temperatures, typically in the range of −70 °C, to create a reversible conduction block in targeted peripheral nerves [7,8,9]. Unlike more traditional neurolytic procedures that rely on heat or chemical agents, cryoneurolysis preserves crucial connective tissues (e.g., the perineurium and epineurium), thereby reducing the likelihood of permanent damage while still providing meaningful analgesic effects. Cryoneurolysis involves the use of a highly pressurized cooling gas (nitrous oxide or carbon dioxide) circulating through a sealed probe. As the gas expands, pressure drops, decreasing the temperature and creating an ice ball at the tip, due to the Joule-Thomson effect [10]. Several factors influence the degree of analgesia obtainable with cryoneurolysis, specifically:The distance between the probe and the target nerve;The cryoprobe diameter;The size of the resulting ice ball;The temperature of the immediately surrounding tissue (such as blood, which acts as a heat sink);The rate and duration of cold application. The latter two factors are highly dependent upon the gas flow rate and the number of ‘freeze cycles’ applied, usually with 2–3 min of freezing followed by 0.5–2 min of thawing [11].

Lowering the temperature to more extreme values creates irreversible tissue damage; this application is called cryoablation [11]. Cryoablation and cryoneurolysis are different interventions with different mechanisms of action. Cryoablation involves the application of liquid nitrogen (reaching temperatures of −140 °C) to obtain irreversible tissue damage; cryoneurolysis reaches higher temperatures (−70 °C), allowing nerves to regenerate. In our search, we found that authors often consider these two terms as synonyms, both indicating a reversible nerve conduction block obtained with the application of cold temperatures. To include more studies and obtain a more complete overview of this technique in pain management, we decided to include all studies mentioning cryoablation, cryoneurolysis, or cryoanalgesia. Clinical interest in cryoneurolysis has expanded considerably, fueled by its demonstrated efficacy in managing refractory pain conditions, as well as its favorable safety profile. Indeed, evidence suggests that cryoneurolysis can be applied to both sensory and mixed nerves for various chronic pain syndromes, providing meaningful relief without the risks associated with permanent nerve destruction or high-dose analgesics [7,8]. Additionally, advances in imaging guidance (e.g., ultrasound, fluoroscopy, and computed tomography) have improved the accuracy of nerve localization, further enhancing procedure success and reducing complications. This narrative review aims to explore the evolving role of neuromodulation therapies, with a particular focus on cryoneurolysis and its potential to revolutionize chronic pain treatment. By examining principles, mechanisms, and clinical outcomes of cryoneurolysis across various chronic pain conditions, we hope to shed light on emerging best practices and inform the development of safer, more effective, and personalized interventions for individuals living with persistent pain.

## 2. Materials and Methods

We conducted a systematic search of PubMed, Embase, Web of Science, and the Cochrane Library for studies published between 2010 and 2025 examining the effects of cryoanalgesia on chronic pain. The research was conducted independently by all the authors. Search terms included “Cryoneurolysis,” “Cryoanalgesia,” “Cryoablation,” and “Chronic pain”; Boolean operators (AND, OR) were used to combine terms. The last search was conducted on 31 March 2025. We searched clinicaltrial.gov and EUDRACT for trials in progress on humans. We included randomized controlled trials, cohort studies, and case reports that evaluated the effects of Cryoanalgesia on Chronic pain. Papers were excluded from the analysis if they met one of the following criteria: -Papers describing the application of cryoneurolysis during a surgical procedure (i.e., intraoperative application on the intercostal nerves for post-thoracotomy pain).-Papers which did not report a statistical analysis (apart from case reports) or simply described a study protocol.-Reviews and all studies not including novel data.-Papers written in a non-English language.

Papers were searched and screened according to PRISMA guidelines.

## 3. Results

We present the number of identified articles, those screened, and those that fulfilled inclusion and exclusion criteria in the PRISMA flow chart.

Figure 1 reports the PRISMA Flow chart with included and excluded studies.

After removing duplicates and excluding non-relevant papers, 55 studies were identified within our search parameters.

In total, 4 of these studies were RCTs, 16 were retrospective, 4 were prospective observational studies, and 31 were case reports or case series with less than 5 patients studied.

The included studies had very heterogeneous inclusion criteria (not all authors reported having performed a test block before cryoneurolysis, nociceptive and neuropathic pain were not differentiated, and different nerves were targeted).

Different devices were used with different protocols of treatment, but not all authors reported details of the temperature and number of cycles performed.

Targets were different as studies focused on different anatomical regions, and various nerves were treated.

Figure 2 represents anatomical regions investigated by different authors, intercostal nerves are the most studied targets, particularly for post thoracotomy chronic pain.

The characteristics of the included studies are summarized in Table 1.

In order to better define the effectiveness of cryoneurolysis, we narrowed our analysis on studies reporting a 6-month follow-up or longer and including a clear definition of meaningful pain relief (>50% from baseline). We stated the outcome as positive if more than 50% of patients reported more than 50% pain relief from baseline.

Table 2 shows the studies fulfilling these criteria.

## 4. Discussion

Despite the major interest in cryoneurolysis by pain physicians, the published evidence is modest.

Our analysis included four RCTs, with different anatomical targets, only two of which reported a positive outcome with a better response in the treatment group versus the comparison (sham or steroid injections).

Regarding retrospective and prospective studies, the results were overall stated as successful; however, most authors did not use a cutoff to define the success of the procedure, except for six studies who reported >50% pain relief at different follow up (generally less than six months), and one study with a success rate of 8% with a cutoff of only 30% pain relief [20,23,27,28,29,31,32].

Most authors reported only a mean decrease in NRS scores, ranging from 2 to 4.2 points of reduction.

All case reports and case series reported positive results, often with 100% pain relief.

Given the great heterogeneity of the included studies, it is not possible to draw a definite conclusion about the efficacy of cryoneurolysis; however, the published evidence indicates a favorable outcome for this technique, at least in the short term, with an excellent safety profile.

Narrowing the analysis on studies with a follow up of at least 6 months or more we found 16 studies, of which 9 were case reports or case series with less than 5 patients treated (Table 2).

No recommendations can be given regarding the best anatomical targets, since studies are very heterogeneous regarding targeted nerves. The best results were reported by the RCT of Radnovich et al. in 2017 [12]. The authors applied cryoneurolysis on the saphenous nerve, reporting a successful outcome at 4 months in more than 80% of patients. The saphenous nerve has been targeted (in its infrapatellar branch) by McLean, who reported more than 50% pain relief in all 23 patients treated, even if only 4 of them reached a 6-month follow-up [31].

A recent large study targeting the genicular nerves confirmed the positive outcomes associated with cryoneurolysis, even if the percentage of patients with >50% pain relief was reported only at a three-month follow-up [35].

Cryoneurolysis application on the occipital, sciatic, and medial branch nerves appears not to be supported by evidence since three RCTs reported no difference versus sham [13,14,15].

It is worth noticing that despite the poor evidence of efficacy for cryoneurolysis on chronic pain, several studies reported good outcomes for the application of the same technique for acute pain management, particularly for intercostal cryoablation to manage post thoracotomy acute pain.

Multiple reasons for this difference could be hypothesized, a more precise application on target nerves (given the direct visualization of nerves during thoracotomy or thoracoscopy), a more specific relation between the targeted nerves and the pain condition, a different pathophysiology of pain (marked central sensitization as in long lasting chronic pain could reduce the efficacy of peripherally focused neuromodulation techniques).

There are no reported comparisons of cryoablation with other interventional techniques, such as radiofrequency ablation, except for one RCT published in 2024. The authors did not report any improvement with cryoneurolysis compared to RF or placebo [13].

Even if this comparison is outside the scopes of this review, it must be observed that published evidence for radiofrequency ablation indicate generally more favorable outcomes for RF denervation in all the anatomical district where cryoneurolysis has been applied to [66,67,68].

For example, RF application for chronic knee, sacroiliac, or shoulder pain can provide sustained pain relief in more than 50% of patients up to 12 months; such results have not been observed with cryoneurolysis [66,67,68].

In our review, only three papers compared cryoneurolysis with RF application: one RCT (comparing cryo with Sham and RF) and two observational retrospective studies [13,29,45]. All of these studies reported better results with RF ablation than with cryoneurolysis.

Interesting results have been reported with the application of cryoneurolysis for neuropathic pain. Yoon et al. enrolled 22 patients for cryoneurolysis of refractory peripheral neuropathy on various nerves and showed significant mean reductions in VAS scores for pain over a 12-month period, maintaining a 50% pain relief at 6-month follow-up [18]. Another study reported the effects of cryoneurolysis on three patients with recalcitrant neuropathic anterior femoral cutaneous nerve pain [38]. Pain relief of more than 80% was maintained 1 year after the intervention. Lastly, a case report involving a refractory sural neuroma, with severe pain refractory to medications and repeated surgeries, reported significant and sustained pain relief after cryoablation [58]. The overall quality of the published evidence is very low, more than half of the included studies are case reports or case series, which are prone to publication bias (reporting only cases with positive outcome).

Most papers did not describe the protocol of cryoneurolysis used, since duration of application and temperature target are two critical factors in obtaining analgesia; this aspect is a serious limit to the quality of the evidence.

Regarding adverse events, cryoneurolysis can be considered a safe procedure. There are no reports of permanent nerve damage. The only contraindications are Raynaud syndrome, cryoglobulinemia, and cold urticaria [10]. The risks of cryoneurolysis are similar to other needle-based percutaneous procedures and include bleeding, bruising, and infection. Specific risks include permanent injury to the nerve, injury to the surrounding tissue, and discoloration of the skin if the cannula is retracted prior to resolving the ice ball and allowing contact with other areas near the target site [10].

Myonecrosis after cryoneurolysis for knee pain has been reported. It is not clear whether this complication was related to the procedure itself (the accidental transmission of nitrous oxide as a cooling gas into the tissue could increase the risk of infection) or to the transmission of skin bacteria into deep tissue during the procedure [69,70].

## 5. Conclusions

Cryoneurolysis is a safe technique, only few adverse events, mostly mild and self-limiting, have been reported. Its efficacy in chronic pain has been demonstrated only in the short-term (less than 6 months) follow-up for various targets (saphenous nerve, sciatic, intercostal, suprascapular).

Given the paucity and low quality of the studies, it is not possible to give recommendations for the application of cryoneurolysis for chronic pain and it cannot be considered a first line treatment.

More robust and well conducted studies are needed, to assess the intrinsic efficacy of cryoneurolysis and to compare it to existing technologies, such as radiofrequency ablation.

We suggest considering these aspects in future study designs:-Precise protocol description for cryoneurolysis application (temperature, duration of application, number of freezing cycles used);-Specific neural targets;-Specific patient population (i.e., chronic post-surgical pain after total joint replacement).

## Figures and Tables

**Figure 1 medicina-61-01090-f001:**
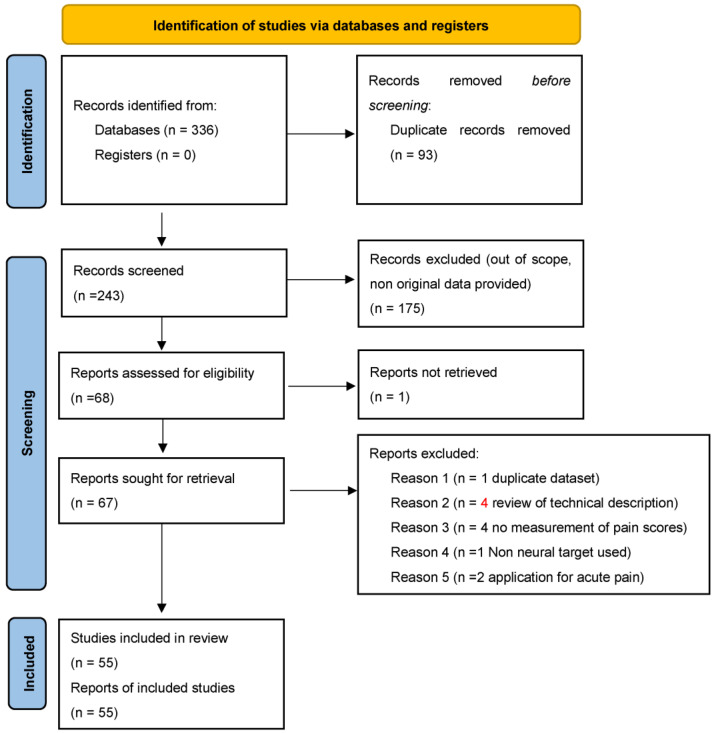
Prisma flowchart.

**Figure 2 medicina-61-01090-f002:**
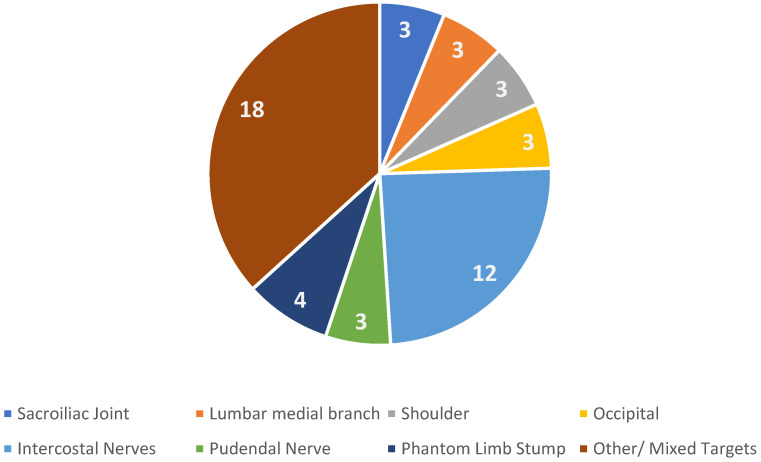
Number of studies for each anatomical target.

**Table 1 medicina-61-01090-t001:** Characteristics of included studies.

Author, Year	Design	Patients	Targets	Guidance	Follow Up	Outcomes	Notes
Radnovich, 2017 [12]	RCT vs. Sham	180	Saphenous Nerve	Landmark	120 days	87.5% responders at 120 days (VAS reduction > 50%)	61.2% of responders in the sham group at 120 days. No adverse events
Truong, 2024 [13]	RCT vs. Sham and RF	120	Lumbar medial branch	Fluoroscopic	6 months	No improvement compared to RF or Placebo	
Ilfeld, 2023 [14]	RCT vs. Sham	144	Sciatic (post-amputation pain)	US	4 months	No improvement compared to Placebo	
Kvarstein, 2019 [15]	RCT vs. Steroid Injection	52	Occipital Nerve	US	18 weeks	Pain relief > 50% in 35% of patients (no difference)	
Grigsby, 2021 [16]	OP	26	Occipital Nerve	Blind	56 days	NRS reduction > 2 points in 35% of patients	
Perry, 2022 [17]	OP	40	Sural, Saphenous, Superficial, and/or Deep fibular nerves	US	6 months	Mean NRS reduction 2.8 (<than 50% from baseline)	22 patients completed follow-up
Yoon, 2016 [18]	OP	22	Peripheral neuropathy (mixed 3 plantar neuromas, 3 ileoinguinal, 4 posterior tibial, 7 saphenous, 1 gluteal, 1 sural, 1 genicular, 2 digital nerves)	US	12 months	3.2 mean VAS reduction at 12 months. Mean pain relief > 50% at 3 months	
Zhan, 2020 [19]	OP	18	Intercostal	CT	12 months	2.3 and 1.3 mean VAS reduction at 6 and 9 months, respectively	No significant reduction at 12 months
Das, 2023 [20]	OR	83	Sacroiliac Joint	US	6 months	69% of patients with pain relief of > 50% after 6 months	
Sidebottom, 2011 [21]	OR	17	TMJ (auricular nerve and TMJ capsule)	Surgical	NS	Mean duration of pain relief 7 months; 3/17 patients were pain-free at 12 months	2 temporary numbness
Prologo, 2017 [22]	OR	21	Phantom Limb Neuroma	US	194 ± 99 Days	Mean NRS reduction 4.2	
Yasin, 2019 [23]	OR	13	Intercostal Nerves	CT	Various, 2–18 months	61.5% of patients with pain relief > 50%	1 pneumothorax and 3 pseudohernia
Moore, 2010 [24]	OR	18	Intercostal Nerves	CT	Variable (mean 51 days)	Mean pain relief < 50%	
Wolter, 2011 [25]	OR	91	Lumbar Medial Branch	CT	3 months	Mean NRS decrease 3.50 points	
Kim, 2015 [26]	OR	38	Occipital Nerve	Landmark	6 months	Mean NRS Improvement 3.8	2 post op neuritis and 1 hematoma
Nemecek, 2023 [27]	OR	24	Various (intercostal, saphenous, peroneal,	US	6 months	Pain reduction > 30% in 2/24 Patients	
Parekattil, 2021 [28]	OR	35	Genitofemoral, ilioinguinal and inferior hypogastric nerve	Blind (surgical)	1 month	Pain reduction > 50% in 68% of patients	
Tinnirello, 2020 [29]	OR	10	Genicular nerves	US	6 months	Pain relief > 50% in 50% of patients at 6 months	Comparison between Cryo, Pulsed, Cooled, and Conventional RF
Nezami, 2022 [30]	OR	14	Intercostobrachial nerve	CT	6 months	Mean NRS decrease 2.9	
McLean, 2020 [31]	OR	23	Saphenous nerve, infrapatellar branch	Blind	Variable	Pain relief > 50% in all patients	Follow up at 6 months for 4 patients, NS for others.
Calixte, 2019 [32]	OR	279	Genitofemoral, ilioinguinal and inferior hypogastric nerve	Blind (Surgical)	5 years	Pain relief > 50% in 64% of patients	
Bellini, 2015 [33]	OR	18	Facet, knee, sacroiliac joint		4 months	Mean NRS decrease 4/10	
Prologo, 2018 [34]	OR	14	Pudendal nerve	CT	18 months	Responder rate 63%	Responder definition not stated
Lo Bianco, 2025 [35]	OR	90	Genicular nerves	US	9 months	Mean NRS 5 ± 1 from 7 ± 2 at Baseline	Pain relief > 50% in 56% of Patients at 3 months (NS at 9 Months)
Filipovski, 2024 [36]	CS	3	Superficial peroneal nerve	US	5 years	2 patients pain-free after 5 years, 1 without results	
Stogicza, 2024 [37]	CS	4	Suprascapular, axillary, lateropectoral nerves	US	6 months	3 patients with pain relief > 60%	
Dalili, 2021 [38]	CS	3	Anterior femoral cutaneous nerve	MRI	12 months	50% VAS reduction at 12 months in all patients	No adverse events
Moesker, 2014 [39]	CS	5	Phantom limb	US	5–30 months	60% of patients with > 50% pain relief	No adverse events
Sahoo, 2021 [40]	CS	5	Lateral branches of sacral dorsal nerve roots	US and Fluoro	6 months	100% of patients with >50% pain relief	
Mendes-Andrade, 2024 [41]	CS	2	Sacrococcygeal nerve	Fluoro		>50% pain relief in 100% of patients	
Shaffer, 2022 [42]	CS	3	Digital nerves	US	1 year	Pain relief 100%	
Gabriel, 2024 [43]	CR	1	Intercostal nerves	US			
Cachemaille, 2023 [44]	CS	4	Alveolar nerves	Blind	3 months	Pain relief > 50% in 2 patients	
Kocân, 2022 [45]	CS	2	Lumbar medial branches	Fluoroscopic	6 months	Pain relief 50% at 3 months (<50% at 6 months, better result with RF)	1 patient treated with RF, 2 with cryo
Connelly, 2013 [46]	CS	3	Intercostal nerves	US	Variable	Pain relief > 50% in 2 patients for 9 months, in 1 patient for 3 months	
Matelich, 2022 [47]	CS	3	Suprascapular nerve	US	3–6 months	Duration of pain relief: 3–6 months	Pain scores not recorded
Fox, 2019 [48]	CS	3	Pudendal nerve	CT	NS	2 patients with > 50% pain relief	
Sarridou, 2022 [49]	CR	1	Stellate ganglion	US	6 months	Pain relief > 50% for 6 months	
Ramsook, 2016 [50]	CR	1	Hip stump neuroma	US	NS	Pain relief	
Joshi, 2017 [51]	CR	1	Posterior femoral cutaneous nerve	MRI	5 months	Pain relief 100% at 6 months	
Kalava, 2024 [52]	CR	1	Intercostal nerves	CT	3 months	100% pain relief	
Sen, 2023 [53]	CR	1	Intercostal nerves	US	5 days	Pain relief 100%	
Jung, 2024 [54]	CR	1	Intercostal nerves	Surgical	8 weeks	Pain relief > 50%	
MacRae, 2023 [55]	CR	1	Superficial fibular nerve	US	5 months	Pain relief (no NRS reported)	
Perese, 2022 [56]	CR	1	Intercostal nerves	US	2 months	Pain relief > 50%	
Koethe, 2014 [57]	CR	1	Intercostal nerves	CT	8 weeks	Paine relief > 50%	
Rhame, 2011 [58]	CR	1	Sural nerve	NS	3 months	“Excellent pain relief”	
MacRae, 2023 [59]	CR	1	Suprascapular nerve	US	7 months	NRS < 2	
Weber, 2019 [60]	CR	1	Intercostobrachial nerve	Blind	1 month	Pain relief > 50%	
Yarmohammadi, 2011 [61]	CR	1	Celiac plexus	CT	6 months	Pain relief 70%	
Rupp, 2022 [62]	CR	1	Suprascapular nerve	US	3 months	Pain relief > 50%	
Kalava, 2024 [63]	CR	1	Intercostal nerves	US	2 months	Pain relief > 50%	
Hampton, 2023 [64]	CR	2	Pudendal nerve	Blind	NS	Pain relief for 3–4 Months	NRS not measured
Fiala, 2022 [65]	CR	1	Phantom limb stump	US	NS	Pain relief < 6 weeks	
Gabriel, 2024 [43]	CR	1	Intercostal nerves	US	6 months	Pain relief > 50%	

RCT: Randomized Clinical Trial; OR: observational retrospective; OP: observational prospective; CS: case series; CR: case report; US: ultrasound; NS: Not Stated; MRI: Magnetic Resonance Imaging, VAS: Visual Analog Score, NRS: Numerical Rating Score, CT: Computerized Tomography.

**Table 2 medicina-61-01090-t002:** Studies with a follow-up of 6 months or more and a definition of meaningful pain relief. Positive outcome is considered pain relief of more than 50% in more than 50% of patients.

Author, Year.	Patients	Targets	Guidance	Positive Outcome at 6 Months
Truong, 2024 [13]	120	Lumbar medial branch	Fluoroscopic	NO
Perry, 2022 [17]	40	sural, saphenous, superficial, and/or deep fibular nerves	US	NO
Yoon, 2016 [18]	22	Peripheral neuropathy (mixed 3 plantar neuromas, 3 ileoinguinal, 4 posterior tibial, 7 saphenous, 1 gluteal, 1 sural, 1 genicular, 2 digital nerves)	US	NO
Das, 2023 [20]	83	Sacroiliac joint	US	YES
Tinnirello, 2020 [29]	10	Genicular nerves	US	YES
McLean, 2020 [31]	23	Saphenous nerve, Infrapatellar branch	Blind	YES
Calixte, 2019 [32]	279	Genitofemoral, ilioinguinal and inferior hypogastric	Surgical	YES
Stogicza, 2024 [37]	4	Suprascapular, axillary, lateropectoral	US	YES
Dalili, 2021 [38]	3	Anterior femoral cutaneous nerve	MRI	YES
Moesker, 2014 [39]	5	Phantom limb	US	YES
Sahoo, 2021 [40]	5	Lateral branches of sacral dorsal nerve roots	US and Fluoroscopic	YES
Shaffer, 2022 [42]	3	Digital nerves	US	YES
Kocân, 2022 [45]	2	Lumbar medial branches	Fluoroscopic	NO
Connelly, 2013 [46]	3	Intercostal	US	YES
Sarridou, 2022 [49]	1	Stellate ganglion	US	YES
Yarmohammadi, 2011 [61]	1	Celiac plexus	CT	YES

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
