# Peer review of "Innovations in Chronic Pain Treatment: A Narrative Review on the Role of Cryoneurolysis"

_medicina, 2025, doi:10.3390/medicina61061090_

Round 1
Reviewer 1 Report
Comments and Suggestions for Authors
A major concern with this manuscript is the inconsistent and imprecise use of terminology throughout the whole manuscript. While the title clearly indicates that the review focuses on “cryoablation” the majority of included studies actually pertain to cryoanalgesia or cryoneurolysis, which differ significantly from cryoablation in both mechanism, pathophysiology and clinical purposes. Cryoablation involves irreversible tissue destruction, whereas cryoneurolysis represents a reversible nerve conduction block. This difference is not adequately addressed in the manuscript, it causes confusion and undermines the clarity and focus of the paper.
Here are additional specific suggestions for improvement:
- As mentioned, the Introduction is insufficient because the difference between cryoneurolysis and cryoablation is not clear. The pathophysiology is completely different. This is the key, please revise to provide better distinguishing between the two. The last paragraph of the introduction section (and much of other introductory content) is about cryoneurolysis and the title of the paper states the paper is about cryoablation?
- There are formatting inconsistencies throughout the manuscript. Please download the standard MDPI template and revise the manuscript according to it. Examples: references should be in square brackets, „INTRODUCTION“ does not need to be all in caps, reference list needs to be in MDPI style, tables and figures are not formatted properly and why is the Disclaimer/Publisher’s Note at the end of the document deleted?
- According to your PRISMA diagram, the number of included studies should be 54, not 55. Please double-check if the numbers are correct.
- Tables 1 and 2 need to be cited in the text before they appear in the document.
- Inclusion and exclusion criteria need to be clearly stated.
- Was the search conducted independently by multiple reviewers?
- The phrase "we checked clinicaltrial.gov …" should be revised to a more formal academic expression (searched, for example)
- Quality of evidence is not assessed or discussed in results nor in discussion section. Given that over half of the included studies are case reports or small case series, the strength of the conclusions drawn must be discussed even briefly.
- Discussion section lacks comparative discussion with other interventional techniques. What about radiofrequency ablation?
- Limitations? The authors did not discuss the potential for publication bias, considering the predominance of case reports with overwhelmingly positive outcomes..
- The conclusion that cryoablation is “safe and promising” is too strong of a statement. While cryoablation may be beneficial in selected cases, strong claims that are not supported by high-quality comparative trials should be avoided.
Author Response
REVIEWERS' COMMENTS A major concern with this manuscript is the inconsistent and imprecise use of terminology throughout the whole manuscript. While the title clearly indicates that the review focuses on “cryoablation” the majority of included studies actually pertain to cryoanalgesia or cryoneurolysis, which differ significantly from cryoablation in both mechanism, pathophysiology and clinical purposes. Cryoablation involves irreversible tissue destruction, whereas cryoneurolysis represents a reversible nerve conduction block. This difference is not adequately addressed in the manuscript, it causes confusion and undermines the clarity and focus of the paper.
Here are additional specific suggestions for improvement:
- As mentioned, the Introduction is insufficient because the difference between cryoneurolysis and cryoablation is not clear. The pathophysiology is completely different. This is the key, please revise to provide better distinguishing between the two. The last paragraph of the introduction section (and much of other introductory content) is about cryoneurolysis and the title of the paper states the paper is about cryoablation? Thanks for this observation. While it’s technically true what you say it must be noticed that in the cited papers the terms cryoblation, cryoanalgesia and cryoneurolysis are used as sinonyms so we decided to include all of these terms in our search and review. We included a detailed explanation in the introduction to clarify this issue. In order not to create misunderstandings we changed also the title in “Innovations in Chronic Pain Treatment: A narrative review on The Role of Cryoneurolysis” and referred to cryoneurolysis in our paper
- There are formatting inconsistencies throughout the manuscript. Please download the standard MDPI template and revise the manuscript according to it. Examples: references should be in square brackets, „INTRODUCTION“ does not need to be all in caps, reference list needs to be in MDPI style, tables and figures are not formatted properly and why is the Disclaimer/Publisher’s Note at the end of the document deleted?. Thanks for the observation, we updated the paper with proiper formatting.
- According to your PRISMA diagram, the number of included studies should be 54, not 55. Please double-check if the numbers are correct.
Thanks for noticing, we found a typing mistake and we fixed it
- Tables 1 and 2 need to be cited in the text before they appear in the document.
- Inclusion and exclusion criteria need to be clearly stated.Thanks for your comment, we re-wrote that section better stating exclusion criteria
- Was the search conducted independently by multiple reviewers?Thanks for the observation, we added this information in the Method Section
- The phrase "we checked clinicaltrial.gov …" should be revised to a more formal academic expression (searched, for example) We changed the sentence
- Quality of evidence is not assessed or discussed in results nor in discussion section. Given that over half of the included studies are case reports or small case series, the strength of the conclusions drawn must be discussed even briefly. We expannded the discussion section highlighting the quality of the evidence
- Discussion section lacks comparative discussion with other interventional techniques. What about radiofrequency ablation? Thanks for your comment. Comparing Cryo with other techniques is outside the scope of the review, however, a brief comparison with radiofrequency ablation has been mentioned and we expanded that section
- Limitations? The authors did not discuss the potential for publication bias, considering the predominance of case reports with overwhelmingly positive outcomes. Thanks for your comment, we mentioned these biases
- The conclusion that cryoablation is “safe and promising” is too strong of a statement. While cryoablation may be beneficial in selected cases, strong claims that are not supported by high-quality comparative trials should be avoided. Thanks for your comment We believe that even if the overall quality of the evidence is poor the safety of the technique is undeniable (since few minor adverse events have been reported), we rewrote the sentence highlighting the need for better evidence
Reviewer 2 Report
Comments and Suggestions for Authors
Dear authors
I reviewed an excellent paper on chronic pain, I congratulate you.
Introduction: a good perspective on chronic pain and on cryoneurolysis as a therapeutic approach for chronic pain with failure of conservative methods. The multiple approaches on chronic pain include pharmacologic and non-pharmacologic approaches, with a large spectrum. ALso, conservative and interventional approaches are used, as chronic is a versatile issue. I found appropriate to mention some innovative techniques in the chronic pain management, as for instance https://doi.org/10.3390/biomedicines11071888 as a pharmacological approach.
Material and methods: a systematic review on multiple databases. Papers were screened with PRISMA criteria. An important number of 55 trials remained in the analysis.
Results were properly displayed in a table with the main features.
Conclusions. The effectivenes of cryoablation is evident in the research. However, the trials were inhomogeneous, and a proper recommendation could not be done. A perpsective on the available literature revealed the necessity of continuing research in this field.
Author Response
REVIEWERS' COMMENTS
Dear authors
I reviewed an excellent paper on chronic pain, I congratulate you.
Introduction: a good perspective on chronic pain and on cryoneurolysis as a therapeutic approach for chronic pain with failure of conservative methods. The multiple approaches on chronic pain include pharmacologic and non-pharmacologic approaches, with a large spectrum. ALso, conservative and interventional approaches are used, as chronic is a versatile issue. I found appropriate to mention some innovative techniques in the chronic pain management, as for instance https://doi.org/10.3390/biomedicines11071888 as a pharmacological approach.
Thanks for your comment, we added the suggested reference to the paper
Material and methods: a systematic review on multiple databases. Papers were screened with PRISMA criteria. An important number of 55 trials remained in the analysis. Thanks for your comment,
Results were properly displayed in a table with the main features.Thanks for your comment,
Conclusions. The effectivenes of cryoablation is evident in the research. However, the trials were inhomogeneous, and a proper recommendation could not be done. A perpsective on the available literature revealed the necessity of continuing research in this field.
Thanks for your comment, we expanded the conclusion and discussion section
Round 2
Reviewer 1 Report
Comments and Suggestions for Authors
Thank you for addressing all my comments and improving the overall quality of the paper. I have no further suggestions at this time.